

# Geographic recommender systems in e-commerce based on population

Mohamed Shili[1] and Osama Sohaib[2,3]

[1] Innov'COM Laboratory, National Engineering School of Carthage, University of Carthage, Carthage, Tunisia
[2] School of Business, American University of Ras al Khaimah, Ras al Khaimah, United Arab Emirates
[3] School of Computer Science, University of Technology Sydney, Sydney, Australia

## ABSTRACT

Technological advancements have significantly enhanced e-commerce, helping customers find the best products. One key development is recommendation systems, which personalize the shopping experience and boost sales. This paper explores a novel geographic recommendation system that uses demographic data, such as population density, age, and income, to refine recommendations. By integrating geographic and demographic information, like the population size of a country, businesses can tailor their offerings to regional preferences. This targeted approach aims to make recommendations more relevant by considering the behaviors and needs of different geographic areas. We sourced population data from The National Institute of Statistics (Tunisia, INS). This approach improves the importance of product recommendations for particular locations by customizing them based on demographic and geographic measures. The technique creates a better context-aware recommendation system that boosts customer happiness and business proceeds by fusing consumer behavior with extensive demographic data. The method also includes a mathematical model that considers population intensity to refine further recommendations established on the regional model.

# INTRODUCTION

Today, e-commerce plays an important role and has spread everywhere around the globe, as customizing the consumer experience has grown essential to allure and retain customers (*Cheng et al., 2012*; *Jiao, Liu & Xie, 2022*; *Mohamed et al., 2023*). Recommender systems, which suggest products or services based on consumers' preferences and past behaviors, are at the heart of this personalization. However, traditional approaches to recommender systems may need more accuracy when considering geographic and demographic change (*Daud, Mulyanto & Pakaya, 2021*). This is reflected in our work, which focuses on population size for recommendation. Recommendation systems must provide users with relevant products based on their preferences and behaviors. The complexity of modern user needs, specifically geographic location, may be partially captured by traditional recommendation methods (*Khodizadeh-Nahari et al., 2021*). Integrating demographic and geographic data into the recommendation proceeding is a new approach offered by a population-based geographic recommendation system (*Mali*

Corresponding author
Osama Sohaib,
Osama.Sohaib@uts.edu.au

& Rachmawati, 2022). This system can provide more context-based and appropriate recommendations by weighing users' location and local population characteristics (Aditia, 2020). The optimization of marketing and logistics strategies for e-commerce businesses is not only improved by this approach but also improves user satisfaction and engagement.

Recommender systems in e-commerce have traditionally relied on methods established on customer preferences and purchase history. However, these approaches have proven limits in catching contextual data such as consumers' geographic location. Recent research, including Mauro et al. (2022) and Zarindast & Wood (2021), show that including spatial information in recommender systems boosts the appropriateness of suggestions, especially in local market conditions. These studies have shown that consumers are more disposed to be involved with recommendations based on their geographic location, boosting customer happiness and business conversion rates. Various research studies, such as Sharma, Shakya & Marriboyina (2020), Honarparvar et al. (2019), Torres-Ruiz et al. (2023), and Pohan et al. (2023), have illustrated that firms may understand and answer consumers' local rules by integrating geographic information systems (GIS) from recommender systems. These systems permit personalized suggestions on geographic attributes and enhance stock handling and logistics to find regional requests. For example, Cruz et al. (2022) shows that GIS can adequately forecast local purchase tendencies and change marketing and logistics methods, resulting in better e-commerce firm competition in particular regions.

In addition, demographic and regional information in recommender systems offers novel study and application opportunities in e-commerce. Several recent studies, including Cruz et al. (2022), Dewan et al. (2023), Kacprzyk et al. (2021), Sihotang et al. (2021), Kopsachilis & Vaitis (2021), investigate how artificial intelligence and machine learning may be combined with GIS to improve suggestions in real-time. This allows for scrutinizing problematic factors such as population immigration, switching in local preferences, and the evolution of temporal purchase habits. These developments present significant opportunities to improve current recommender systems, and our study contributes to this field by existing a unique technique that incorporates large-scale demographic data with spatial data to offer suggestions suitable to particular market categories.

Moreover, our proposal would contain a user-friendly interface to assist platform administrators in managing and analyzing geographical data for continuous system optimization. By investing in this cutting-edge technology, businesses could enhance customer overhauling and engagement and open up new growth provisions in a progressively concurrent and global e-commerce market. The significant contribution of this paper include:

- The system combines geographic and demographic data to provide recommendations that are more suitable to local preferences, increasing the importance of product recommendations to customers.

- By utilizing demographic and geographic information, businesses can improve a portion of the market and reach potential customers with offers tailored to their needs, making their marketing campaigns longer efficient.
- Businesses may save expenses and delivery time by optimizing inventory gestion and delivery tactics that include user location.
- By providing appropriate goods and services, contextual, location-based suggestions enhance user experience and encourage customer loyalty.
- By presenting bargains appropriate to their zone, the system boosts the potential of conversions by supplying consumers with location-specific deduction and publicity.
- The solution aids the sellers in chopping expenditures related to conveyance by making the best use of delivering routing through geographic data, whatever warrants faster distribution.
- To ease consumer displeasure and boost buying satisfaction, the system ensures that recommendations are restricted to goods accessible in the customer's local zone.
- The system updates customers about popular things in their area by suggesting products presently prevalent in these localities.

The rest of this paper is divided into five sections. 'Literature Review' will be related to our work through a literature review. The proposed architecture is constructed with the necessary components in 'Methodology'. 'Web Application and Experimental Results' concludes with a web application for simulation following the proposed solution. In 'Discussion', we offer a discussion of the results obtained. We present our conclusions and future work in 'Conclusion and Future Work'.

## LITERATURE REVIEW

Recommendation systems and GIS are essential in increasing the achievement of e-commerce systems. Therefore, there was research on expanding various recommendation systems using several methods. Some of them concentrated on customer behavior and geographic position.

In *Nurcahya & Supriyanto (2020)*, the authors proposed a new approach for a content-based recommender system explicitly established for e-commerce platforms, highlighting how adapted product recommendation orders enhance customer experience. This system's capability to personalize suggestions based on singular customer preferences and product features is one of its main advantages because it considerably boosts the recommendations' suitability. In addition to improving customer pleasure, this customization poses platform engagement. Moreover, the system effectively computes similitude among product names and designations, producing pertinent recommendations even when customer interaction information is insufficient. However, the article also explores various disadvantages linked to content-based filtering. The system's efficiency is mainly conditioned on the bore and applicability of product descriptions, which display substantial variance amid distinct goods. Moreover, since it primarily focuses on item grade rather than integrating user behavior or collaborative filtering processes, this approach could not adequately capture the range of customer preferences. This limitation may

result in an overfitting problem, in which the system repeatedly suggests identical items, diminishing the diversity of recommendations and rearing the potential of customer exhaustion.

*Addagarla & Amalanathan (2021)* present a novel product recommendation method for utilizing deep learning methods for visual similarity. The e-SimNet model, as the authors propose, utilizes a convolutional neural network (CNN) called SqueezeNet to abstract visual data from product photos. This allows for the suggestion of clearly linked goods. A fundamental profit of this model is its excellent accuracy of 96.2% with a low fault percentage, the best performer of other well-known models such as Visual Geometry Group (VGG) and residual neural network (ResNet). Integrating deep learning, whatever surmounts the disadvantage of conventional machine learning techniques that often rely on text-based searching, is accredited with this effectiveness.

Moreover, the model is better flexible to customization in picture orientation when data augmentation approaches are used. Smaller e-commerce platforms may be unable to use e-SimNet due to its complex and resource-intensive installation. Moreover, the quality of the input photographs significantly impacts the system's performance, given that low-quality image strength results in erroneous feature extraction. Finally just not less, even when the model is splendid at the production of visible recommendations, it is not entirely narrative for other crucial elements like customer preferred and contextual data, which might enhance the recommendation process even more. E-SimNet is a noteworthy development in visual recommendation systems for online shopping, with impressive remainder among strong accuracy, implementation, and picture quality issues.

In another study *Hussien, Rahma & Wahab (2021)*, the authors examine how essential recommendation systems are to upgrading e-commerce websites. The ability of their systems to provide personalized product recommendations established on singular customer preferences and archival behavior is one of its significant advantages. This characteristic considerably increases consumer satisfaction and boosts fidelity. In addition, recommendation systems can increase selling and transformation rates by successfully guiding customers to products they are likely to buy. By assisting users in navigating extensive product selections, recommendation systems also help reduce the topic of data overload. Nevertheless, the study also discusses particular disadvantages, such as the cold start topic, in which new users or items might not obtain precise recommendations because of insufficient historical data. Moreover, sustaining the performance and accuracy of recommendation algorithms can be problematic as data quantity rises, which could result in complexity and scalability problems.

*Kompan et al. (2022)* analyze why customer tariff preferences and benefit consciousness from recommender systems should be incorporated into a focus on e-commerce. It fills a critical space among the key performance indicators (KPIs) organizations prioritize—revenue and profit—and standard study measures like mean average precision (MAP) and recall. The authors display why this enhancement can simultaneously increase recommendation accuracy and profitability by laying forth a special modification to score-based recommender algorithms. The significant gain of this study is that it can be applicable virtually. It allows a framework that helps e-commerce businesses to

personalize their suggestions corresponding to customer preferences and benefit margins, which advantages both customers and the company. One substantial disadvantage is that laying these strategies into custom could require a greater comprehension of consumer behavior, which might present difficulties for smaller businesses with less data. Furthermore, using past data to train a model can not necessarily consider the current market model, resulting in less valuable suggestions in rapid modification contexts.

*Pleskach et al. (2023)* discuss why machine learning (ML) and artificial intelligence (AI) exist and are utilized to enhance customer participation and decision-making in e-commerce recommender systems. It offers an in-line store-specific microservice architecture that makes context-aware recommendations simple. Enhanced personalization of recommendations established on various contextual features, including period, location, and customer behavior, is one of this approach's profits. It also boosts customer happiness and engagement. However, the paper also underscores difficulties, such as the problem related to the cold start problem, the scalability topic, and the necessity for several clear recommendations. This restriction can obstruct the effectiveness of recommender systems, specifically in adapting to new customers or speedily modifying market conditions. While the proposed solutions provide significant profit in improving e-commerce experiences, they also require a thorough examination of the associated challenges to ensure their successful implementation. Advanced techniques like deep learning and natural language processing also facilitate the system's adaptation to dynamic customer preferences, supplying more appropriate suggestions.

In another study, *Yıldız, Güngör Şen & Işık (2023)* introduced a product recommendation system constructed for fashion retail commerce. The system utilizes customer position information and RFM segmentation to give personalized recommendations. It uses methods like relationship rule extraction and k-means to select whatever things to recommend. It also considers the importance of customized recommendation systems and areas of gaps in the organization of the present study. The proposed approach in the following paper produces bespoke product recommendations using geography and customer fragmentation information. It asks to give customers better accurate and personalized recommendations using complicated algorithms like k-means and relationship rule mining. In addition, measuring the performance of the suggested system contrary to present systems presents essential information about its possible profit. The proposed system's dependence on complex algorithms and consumer information may be one of its drawbacks. This might result from increased processing needs and potential privacy issues using client data for adapted suggestions. Table 1 summarizes the technologies used in GIS personalization and their corresponding outcomes in the field.

## METHODOLOGY

The section explains the approach to drawing up a geographic recommendation system in e-commerce. It incorporates the mathematical formulation and the framework of the proposed method.

**Table 1  Brief of examined models and systems.**

| Paper | Diversity of recommendation | Customer's satisfaction | Time needed for sale | Accuracy of recommendation | Customer preference analysis | Similarity measure efficiency | Using GIS | Data source |
|---|---|---|---|---|---|---|---|---|
| *Nurcahya & Supriyanto (2020)* | Moderate | High | Potentially reduced | High | Limited | Efficient | No | Depend on product designation and names |
| *Addagarla & Amalanathan (2021)* | High | Moderate | No | Yes | Yes | No | No | Utilizes product images with the basic data source |
| *Hussien, Rahma & Wahab (2021)* | Yes | No | Yes | | No | Yes | No | Website online |
| *Kompan et al. (2022)* | Yes | High | Yes | Yes | No | Yes | No | Utilizes feasible datasets for the fashion domain |
| *Pleskach et al. (2023)* | Yes | Yes | Yes | Yes | No | Yes | No | Crucial data sources incorporate customer behavior, demography, and context. |
| *Yıldız, Güngör Şen & Işık (2023)* | Yes | Low | No | No | No | No | No | Not specified |
| Our Proposed System | Yes | Yes | Yes | Yes | Yes | Yes | Yes | INS Tunisia |

## Proposed architecture and mathematical formulation
### Population intensity/geographic balance

Exploiting population density at a product's geographic location is essential this system. The population density and $P_{l_y}$ at the product evaluation site meant by this. The population intensity is standardized by partitioning it by the highest population density $P_{max}$ for each imposing interpreted into consideration to ensure justice and coherence. This normalization ensures that the suggestion method isn't predominant in areas with high population densities.

### Overall average rating

The system includes an overall average rating or bold italic delta to specify the initial or average rating on the condition that items are across the platform have $\delta$. To guarantee that the model is established on the regular behavior seen in the system, this value is utilized as the initiation topic for all customer-product estimation.

### Product/customer biases

Subsequently, establishing the system's overall framework, customer and product, and biases accompanying customers and products are integrated to adapt personal preferences and attributes of the product.

Customer bias $b_x$: Specify a particular customer's tendency to give a product an enhanced or lower rating than the norm.

Product bias $b_y$: Specify a product's propensity to continually score enhanced or worse, irrespective of the person's evaluation.

Their biases change the initial rating to more accurately constitute personal and product-specific variances.

### Mathematical formulation

In this section, we prepare a population-based geographical recommendation modeling mathematically. The fundamental equating incorporates customer and product biases and a geographical balance element established on population intensity, as displayed in Eq. (1) and the example below:

$$\hat{R}_{xy} = \delta + b_x + b_y + \beta * \frac{P_{l_y}}{P_{max}}. \tag{1}$$

- $\hat{R}_{xy}$: It is the classification of customer **x** and product y.
- $\delta$: It parameter means the overall average of grades.
- $b_x$: It is customer bias (constitutes the customer's tendency to give larger or less evaluation).
- $b_y$: This is the second product bias (representing the product's propensity to obtain larger or lower-quality reminding).
- $\beta$: This is a weight criterion for population intensity.
- $P_{l_y}$: The population intensity at the product $l_y$ position geographic.
- $P_{max}$: It is the highest population intensity between all positions regarded (to standardize intensity among 0 and 1).

**Table 2  Influence of population intensity, product bias, and customer bias on projected ratings.**

| Case | $b_x$ (Customer Bias) | $b_y$ (Product Bias) | $P_{l_y}$ (Population Intensity) | $P_{max}$ (Predicted Rating ) |
|------|------|------|------|------|
| 1 | 0.8 | −0.4 | 700 | 4.55 |
| 2 | 1.0 | −0.2 | 1,000 | 5.00 |
| 3 | −0.5 | 0.5 | 1,400 | 4.75 |
| 4 | 0.2 | −0.3 | 500 | 4.35 |
| 5 | −0.1 | 0.0 | 900 | 4.60 |

Let's take an example with real numbers.

$$\delta = 4.5; b_x = 0.8; b_y = -0.4; \beta = 0.7$$

The Intensity of population wherever the product is positioned: $: P_{l_y} = 700/km^2$
The maximal population density in full position $:: P_{max} = 1400/km^2$
Accordingly, the predicted evaluation result is as follows:

$$\hat{R}_{xy} = 4.5 + 0.8 - 0.4 + 0.7 * \frac{700}{1400}$$
$$\hat{R}_{xy} = 4.5 + 0.8 - 0.4 + 0.7 * 0.5$$
$$\hat{R}_{xy} = 4.5 + 0.8 - 0.4 + 0.35$$
$$\hat{R}_{xy} = 4.55.$$

Thus, Eq. (1) illustrates how to apply population intensity to change product suggestions without considering customer preference or concentrated populations in various geographic areas.

### Comparative analysis

This section compares and examines the effects of modification on essential characteristics of the awaited rating, including population intensity and biases. The result of applying several values for these variables is summarized in a table.

$$\delta = 4.5; P_{max} = 1400/km^2; \beta = 0.7.$$

We provide a comparison analysis to aid you in improving see how several factors impact the expected rating. By adjusting these values, we might see how significant characteristics are similar population intensity $P_{l_y}$, customer bias $b_x$, and product bias $b_y$ impact the recommendation outcome $\hat{R}_{xy}$.

The expected ratings $\hat{R}_{xy}$ For the several positions are prepared in Table 2.

The values under which the comparison is made are shown in the next section.

### The proposed architecture

In the following diagram, we show the major components of the proposed architecture of our system, which principally consists of five elements, as shown in Fig. 1 and Table 3. The architecture of the multi-layered e-commerce recommendation system is represented as shown in Fig. 2. Managers may enter a database with personalized recommendations

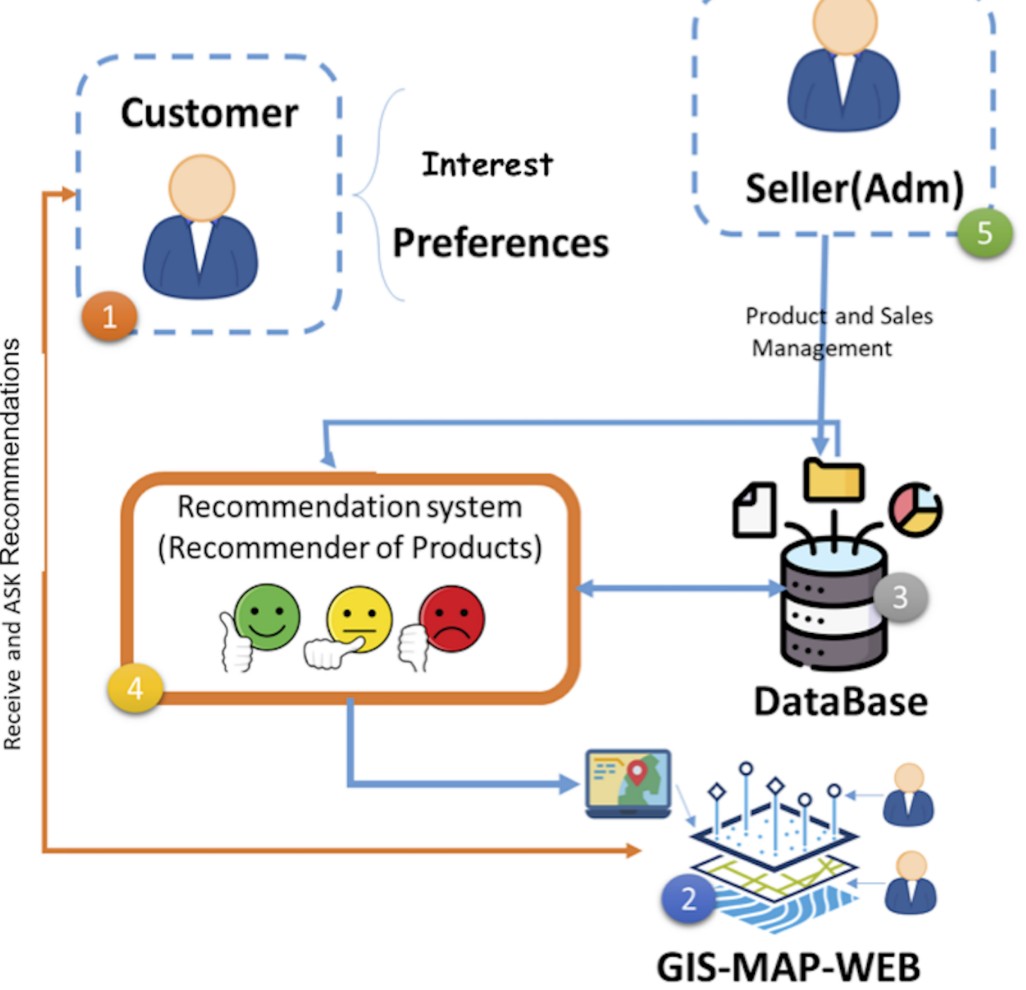

**Figure 1** The proposed architecture for this system.

established by processing consumer preferences and regional data for product management and customer behavior visualization—layer cooperation into the rest to provide the best customer experience and serve specific requests.

## Data collection

The geographic recommendation system draws on various demographic and geographic data sources. These unique characteristics were selected because they directly affect customers' behavior and purchasing ethical values. Specific points were given primacy, including geographical preferences, income levels, and population density. These features have been illustrated in previous studies to impact e-commerce ethical values significantly, and their inclusion enhances the precision and relevancy of product recommendations. Regional differences were described in detail by the National Institute of Statistics (Tunisia, INS), which provided the demographic statistics.

**Table 3  The description of the layers.**

| N | Layers | Components | Description |
|---|--------|-----------|-------------|
| 1 | Customer Layer | - User Profiles<br>- Preference Settings<br>- Behavioral Data | They constitute the customers of the system and this layer represents the situations that inform us of their preferences and choices and is a main context for the geographical recommendation system. |
| 2 | GIS-MAP-WEB Layer | -Geographic Information System (GIS)<br>- Mapping Interface<br>- Geospatial Analysis Tools | This element is in charge of directing geographic data and offering a map interface for visualizing spatial information. It is connected to the database to access customer data and related products according to their preferences |
| 3 | Data Base Layer | - Customer Data<br>- Product Data<br>- Recommendation Data<br>- Transactional Data | This component or element victuals all data, like customer data, products, recommendations released by the system, etc. |
| 4 | Recommendation System Layer | - Recommendation Algorithms<br>- Data Processing Modules<br>- Connection to Database | This element constitutes using the recommendations and algorithms to generate personal recommendations for your customer. It is important to connect to the database to entry the necessity recommendations. |
| 5 | Seller (Adm) Layer | - Administrative Interface<br>- Product Management<br>- Sales Management<br>- Database Management | This element constitutes the system vendors or supervisors on this system and they also use an administrative interface connected to the database to manage products, sales, and other functions accompanying the e-commerce platform. |

In Table 4, we present the sample dataset related to the population. Data preparation, frequently described as data cleaning, is crucial in disposing of this data for analysis. This positioning located and fixed any irregularity, such as exceptions or irregular results, to boost the dataset's quality. This step also solves any inconsistency in the data by standardized variables as necessary. Lacking values is specifically significant; we need to measure the dataset's structure and the type of missing data to handle them well. It is possible to guarantee that the dataset is extensive and prepared for precise and trustworthy analysis by utilizing suitable attribution techniques.

A cluster of geographical features that correlate to population statistics is demonstrated in the Fig. 3 GeoJSON file. Every characteristic in the collection encompasses data on the zone and related characteristic features, including the count of enterprise, employment, and population. Various employments exist for this data, such as mapping, analysis, and visualization.

Figure 4 displays the population distribution mapping, which is essential for geographic investigation because it assists in comprehending the various human–environment interactions and behaviors.

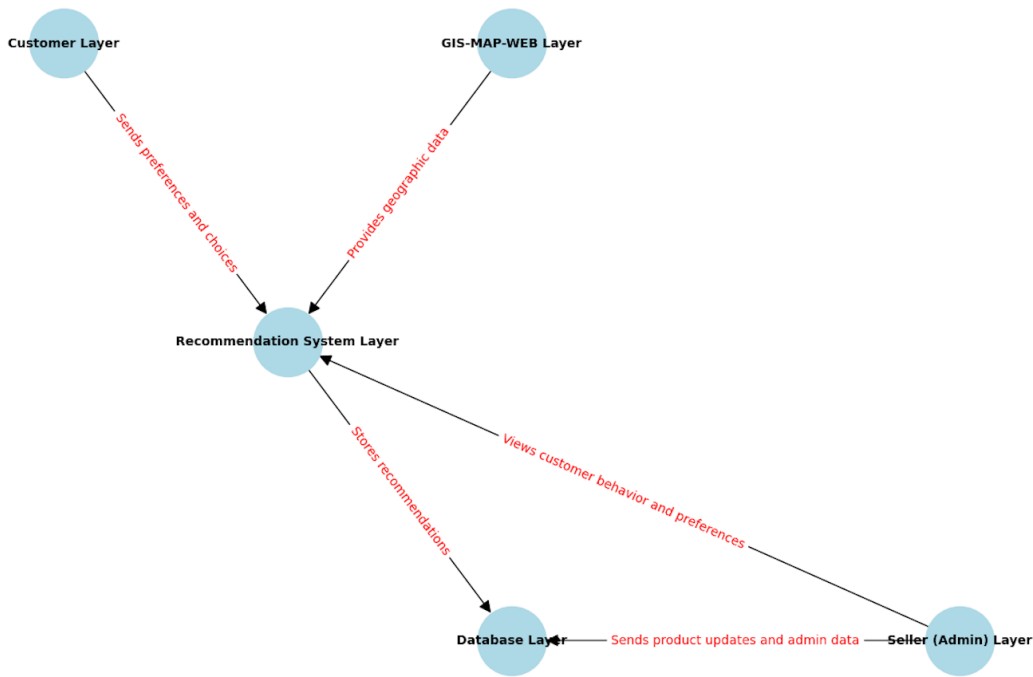

**Figure 2** The system architecture diagram with data flow of the proposed system.

## The preprocessing data

The required proceeding to draw up selling information and demographic data from the INS National Institute of Statistics (Tunisia) for a geographic recommender system is described in the data preparation schema, as shown in Fig. 5. To assure compatibility and importance, it opens with collecting and combining selling and population data, then normalizing and extracting features. Ultimately, the information is disposed of for accurate location-based recommendation *via* data purification and region-specific accumulation.

## Flowchart for proposed approach

The diagram flowchart represents the proposed system's forward action, from information collecting to system upgrade, securing an extensive recommendation system for e-commerce established on geographic information systems and population information. This graph is particularly worthwhile in the original phase of a proposal and assists in determining the high order for improving work. An elaborated flow diagram is a tight-upward view of the action, generally displaying the following steps. The diagram initiates with the starting point, initially by data collection incorporating customer data from direct interactions and demographic data acquired from community databases. This data is then integrated and transformed to extract valuable information. User locations are geocoded, and the data is analyzed to segment customers. Established on the consumer understanding, customized recommendations are produced and presented on a map. The flowchart methodology of this study is accessible in Fig. 6.

**Table 4  Sample dataset of population.**

| ID | Label | POP_5_60 | POP_ADULT | POP_CHOMAG | POP_RURAL | POP_URBAIN | Population |
|----|-------|----------|-----------|------------|-----------|------------|------------|
| 1 | KEBILI SUD | 6,446 | 24,001 | 4,848.202 | 1,6475 | 13,972 | 30,447 |
| 2 | KEBELI NORD | 6,611 | 25,243 | 5,510.5469 | 18,009 | 13,845 | 31,854 |
| 3 | SOUK EL AHED | 6,140 | 21,725 | 4,373.2425 | 8,960 | 18,905 | 27,865 |
| 4 | DOUZ SUD | 3,872 | 14,693 | 2,684.4111 | 8,970 | 9,595 | 18,565 |
| 7 | EL OMRANE SUPERIEUR | 11,808 | 43,705 | 7,067.0985 | 0 | 55,513 | 55,513 |
| 45 | FOUCHANA | 13,117 | 61,751 | 9,009.4709 | 29,761 | 45,107 | 74,868 |
| 46 | MORNAG | 12,135 | 49,383 | 6,503.7411 | 22,990 | 38,528 | 61,518 |
| 75 | EN-NADHOUR | 6,292 | 24,074 | 4,978.5032 | 22,799 | 7,567 | 30,366 |
| 48 | OUED ELLIL | 13,292 | 56,025 | 9,255.33 | 11,466 | 57,851 | 69,317 |
| 49 | MORNAGUIA | 8,408 | 34,279 | 4,445.9863 | 22,853 | 19,834 | 42,687 |
| 50 | BORJ AMRI | 3,519 | 13,889 | 3,100.0248 | 10,889 | 6,519 | 17,408 |
| 51 | DJEDEIDA | 9,005 | 35,743 | 6,183.539 | 16,088 | 28,660 | 44,748 |
| 52 | TEBOURBA | 9,178 | 34,321 | 6,908.8173 | 15,954 | 27,545 | 43,499 |
| 53 | EL BATTANE | 4,055 | 14,922 | 2,656.116 | 12,524 | 6,453 | 18,977 |
| 54 | NABEUL | 14,884 | 58,244 | 5,970.01 | 2,691 | 70,437 | 73,128 |
| 55 | DAR CHAABANE EL FEHRI | 8,670 | 38,111 | 3,399.5012 | 4,641 | 42,140 | 46,781 |
| 56 | BENI KHIAR | 8,630 | 34,502 | 2,425.4906 | 6,450 | 36,682 | 43,132 |
| 57 | KORBA | 14,297 | 54,667 | 4,482.694 | 20,676 | 48,290 | 68,964 |
| 58 | MENZEL TEMIME | 14,588 | 51,057 | 4,120.2999 | 21,264 | 44,381 | 65,645 |
| 59 | EL MIDA | 6,016 | 20,979 | 1,602.7956 | 22,840 | 4,155 | 26,995 |
| 60 | KELIBIA | 12,041 | 46,450 | 5,504.325 | 6,581 | 51,910 | 58,491 |
| 69 | HAMMAMET | 18,475 | 79,380 | 10,335.276 | 24,619 | 73,236 | 97,855 |

## Customer satisfaction and personalization

Figure 7 displays how customization and consumer satisfaction interact with an e-commerce recommendation system. Consumer data input is the initial phase and the basis for collecting customer data. Geographic and demographic information are the two major categories into which this information falls. By tailoring regional preferences and population density, geographic data help customize product recommendations that are requested for a particular locality. Demographic data, such as age, income, and family size, ensure that recommendations match the users' lives and financial capabilities. Customized suggestions result from combining these two data sources, boosting the products' appropriateness. Customers are more satisfied when they are involved with these personalized recommendations, which pose engagement and loyalty. In addition to upgrading the purchasing experience, data-driven personalization also produces permanent relationships with customers and aid in the overall success of the e-commerce platform.

## Recommendation algorithms

The process of a composite recommendation system that incorporates collaborative algorithms and content-based filtering is described as shown in Fig. 8. Customer preferences and item specifics are the first information based on the customer profile data.

```
{
"type": "FeatureCollection",
"name": "pop",
"crs": { "type": "name", "properties": { "name": "urn:ogc:def:crs:EPSG::32632" } },
"features": [
{ "type": "Feature", "properties": { "ID": 1, "C_GOUV": "63", "CODE_DELEG": "6351", "LABEL": "KEBILI SUD", "NB_ENT_PRI": 3387, "NB_LOG": 7759, "POP_5_60": 6446,
"POP_ADULT": 24001, "POP_CHOMAG": 4848.202, "POP_RURAL": 16475, "POP_URBAIN": 13972, "POPULATION": 30447, "TX_CHOMAG": 20.2, "TX_PAUV": 15.8 }, "geometry": { "type":
"MultiPolygon", "coordinates": [ [ [ [ 485725.420456072199158, 3722777.515649601817131 ], [ 485520.394067887158599, 3722731.185670440550894 ],
[ 485348.415329596842639, 3722645.196367065422237 ], [ 485216.124011095729657, 3722512.9050686894916 ], [ 485044.145269447646569, 3722380.613672959152609 ],
[ 484938.312136161781382, 3722301.238935723435134 ], [ 484786.177114196994808, 3722294.624375632964075 ], [ 484673.72952243470354, 3722393.842838686425239 ],
[ 484534.823646973352879, 3722466.603022156283259 ], [ 484395.917662434105296, 3722407.071998563129455 ], [ 484303.313766705454327, 3722281.395268189255148 ],
[ 484276.855474999523722, 3722142.489292409736663 ], [ 484276.855464374297298, 3721990.354273946490139 ], [ 484257.011742668342777, 3721844.833827148191631 ],
[ 484151.178709785628598, 3721778.688130848109722 ], [ 484012.272720962937456, 3721666.240444282535464 ], [ 483926.283397599356249, 3721573.636565097607672 ],
[ 483866.752256264153402, 3721481.032683998346329 ], [ 483774.148258427216206, 3721335.512142159976065 ], [ 483641.856937752687372, 3721196.606173553038388 ],
[ 483509.565619821369182, 3721097.387827319093049 ], [ 483403.73248685739236, 3721037.856701553333551 ], [ 483218.524603417725302, 3720925.409118540585041 ],
[ 483053.160428438568488, 3720773.274111696518958 ], [ 482920.869108996179421, 3720660.826424804050475 ], [ 482729.046661561413202, 3720640.982728267554194 ],
[ 482616.5896886756178, 3720726.972152267117053 ], [ 482590.14079333236441, 3720812.961469599045813 ], [ 482576.911660437297542, 3720918.79459707532078 ],
[ 482490.922251123352908, 3721018.01305977627635 ], [ 482325.558090267295483, 3721051.085924737621099 ], [ 482140.350214703299571, 3721044.471269012894481 ],
[ 482027.902610815886874, 3720984.940344405826181 ], [ 481948.52775373775512, 3720852.648942753206939 ], [ 482014.6734501798637211, 3720740.201340551953763 ],
[ 482047.746199745219201, 3720700.513915459625423 ], [ 481994.82972795702517, 3720607.91003397339955 ], [ 481895.611159594729543, 3720482.233104121871293 ],
[ 481803.007263283827342, 3720376.400084966327995 ], [ 481769.934402461745776, 3720277.181531368754804 ], [ 481756.705254907195922, 3720197.806787384673953 ],
[ 481703.788683956605382, 3720118.431946610566229 ], [ 481631.028502347005997, 3720052.286248301155865 ], [ 481531.809940595179796, 3720012.598933884873986 ],
[ 481465.664230561582372, 3719959.682375910691917 ], [ 481406.133194893365726, 3719953.067810409702361 ], [ 481346.602058672811836, 3719939.838674494996667 ],
[ 481273.84178075636737, 3719919.99496926786378 ], [ 481161.394178848189767, 3719893.536696957889944 ], [ 481042.332006898825057, 3719873.692995701916516 ],
[ 480916.65515650389716, 3719761.245408985763788 ], [ 480804.207546009274665, 3719635.568680277094245 ], [ 480777.749255218543112, 3719536.350125933531672 ],
[ 480751.290967799664941, 3719476.818994217552245 ], [ 480652.07250677770935, 3719450.360820977017283 ], [ 480559.468510029197205, 3719351.14227229822427 ],
[ 480612.385069023701362, 3719284.996563232969493 ], [ 480678.530776046856772, 3719298.225598420482129 ], [ 480784.363806590612512, 3719311.454830223228782 ],
[ 480883.582266535551753, 3719324.683962637558579 ], [ 480962.957105750101618, 3719232.080069587100297 ], [ 480923.269670989946462, 3719099.788663900922984 ],
[ 481002.644510299549438, 3719007.184770772699267 ], [ 481088.633821943425573, 3718934.424488452728838 ], [ 481115.092096565058455, 3718841.820599764585495 ],
[ 481088.633807484409772, 3718762.445756430272013 ], [ 481095.248368081753142, 3718643.383587472140789 ], [ 481115.092073795560282, 3718570.623310635332018 ],
[ 481075.404640118998941, 3718451.561145480256528 ], [ 481095.248341395228636, 3718325.884304760489613 ], [ 481148.164898015209474, 3718226.66584311472252 ],
[ 481253.998019112448674, 3718120.832706518936902 ], [ 481326.758188826905098, 3718034.84338413644582 ], [ 481373.060174440150149, 3717929.010252457272864 ],
[ 481465.664055611239746, 3717836.406357926316559 ], [ 481564.882608575804625, 3717757.031503817532212 ], [ 481637.642888249247335, 3717790.104350314941257 ],
[ 481703.788489196449518, 3717723.958639956079423 ], [ 481796.392375407158397, 3717690.885679936502129 ], [ 481888.996366439096164, 3717717.343954416923225 ],
```

**Figure 3** Examples of data in the GeoJSON format.

Next, content-based filtering analyzes this data using TF-IDF and cosine similarity to compare customer profiles to item attributes and make preparatory recommendations. Equivalent to this, collaborative filtering explores user-item interrelationship data to get similar populations and goods and then improves suggestions established on this history of interactions. In the combined recommendation scores stage, the outcomes of the two filtering techniques are combined, and the scores are incorporated through a weighted average method. The peak products are presented in a catalog of recommended products that the system creates based on the user's interplay behaviors and preferences. This schema allows a prominent model of how various recommendation systems may be incorporated to offer personalized product recommendations.

## Regional analysis algorithms

The recommendation system's regional analysis algorithms are briefly condensed in Fig. 9. The initial phase is named geographic filtering, which adjusts suggestions established on

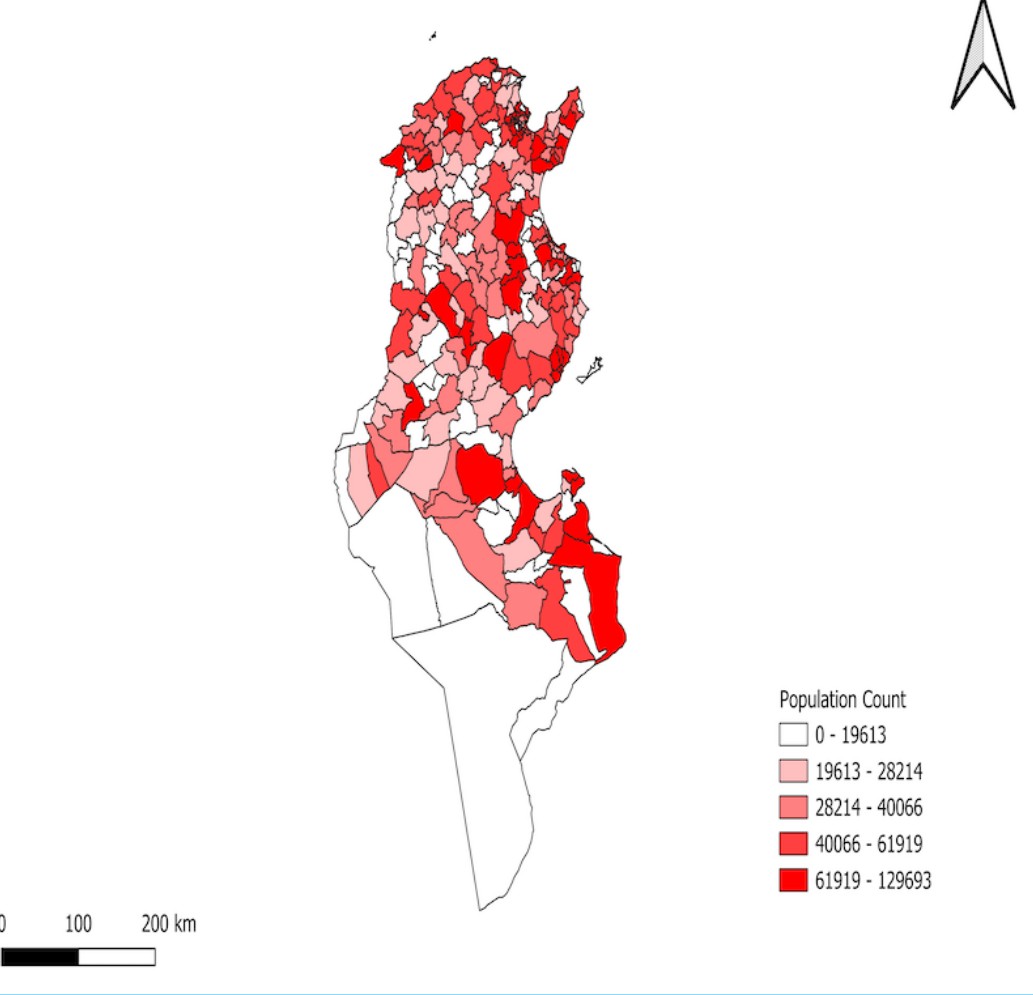

**Figure 4  Population number map.**

the user's position and highlights localized products and agreements. Trend analysis is a helpful tool that improves recommendations by identifying and incorporating popular items and developing trends across various locations. To improve the precision and application of suggestions, understanding from trend and geographic data are combined in the Integrating with recommendation algorithms stage. This phase integrates both of these procedures. Personalized and region-specific product recommendations are on condition that the final product, improved recommendations, warrant that recommendations are contextually appropriate and personalized to the customer's local preferences.

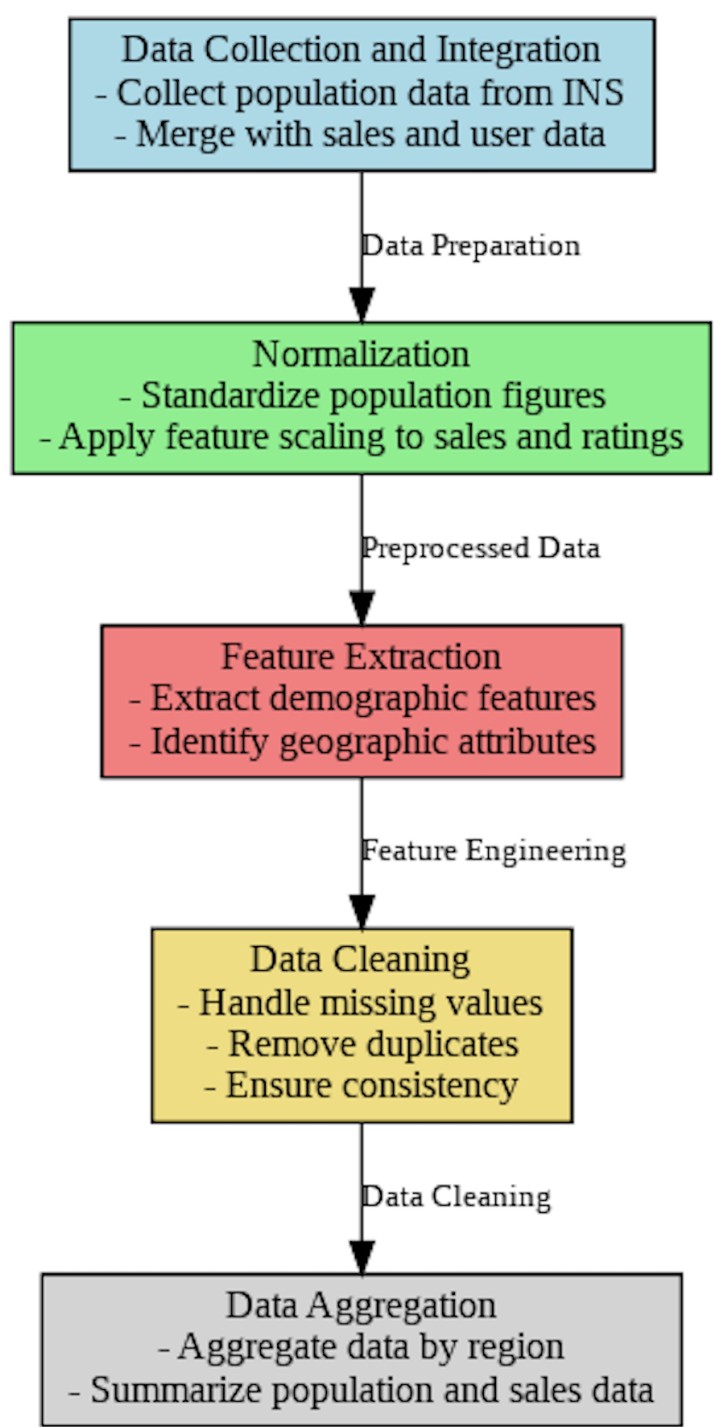

**Figure 5** The preprocessing data of the geographic recommender system.

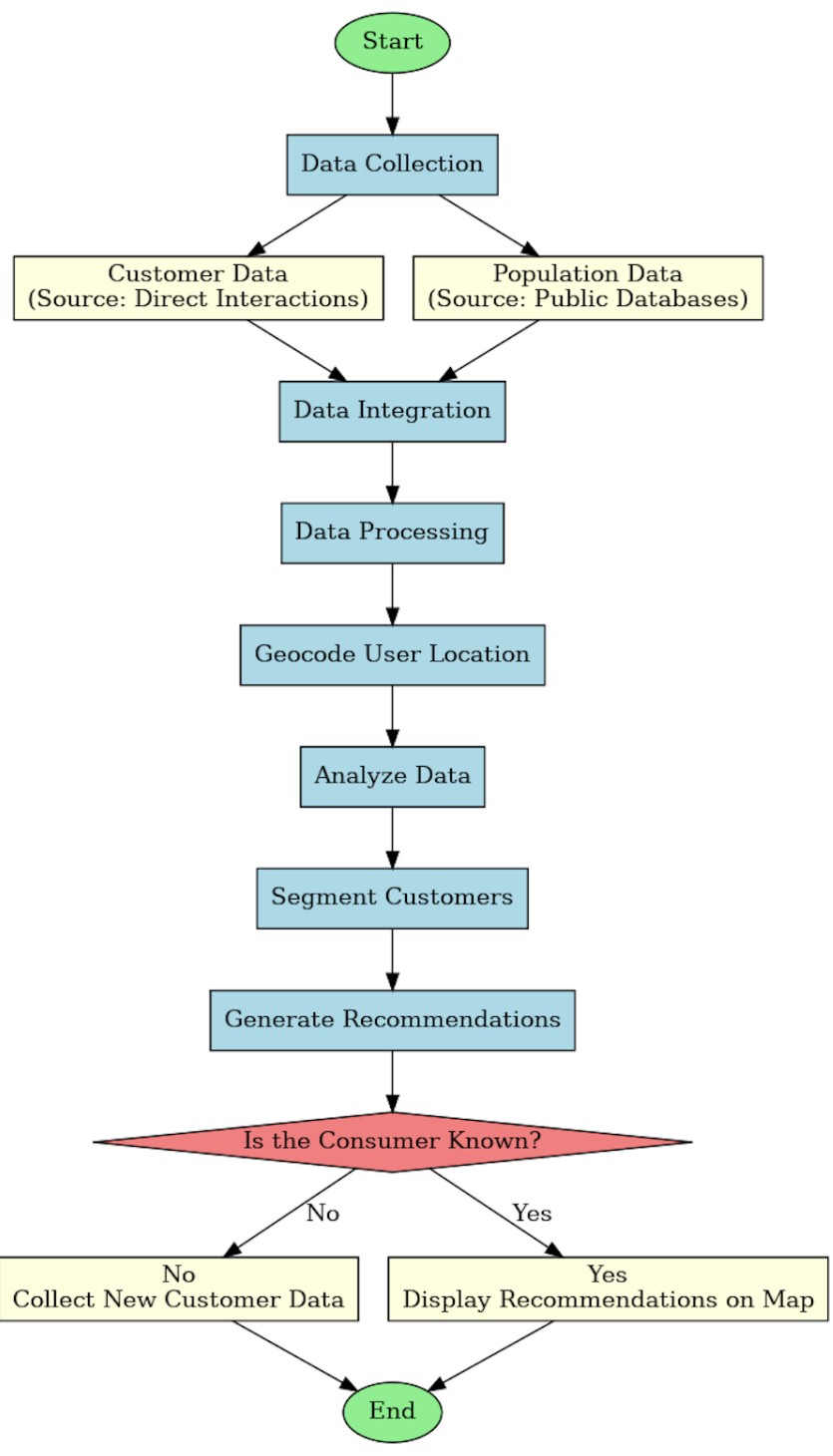

**Figure 6** Flow chart of approach.

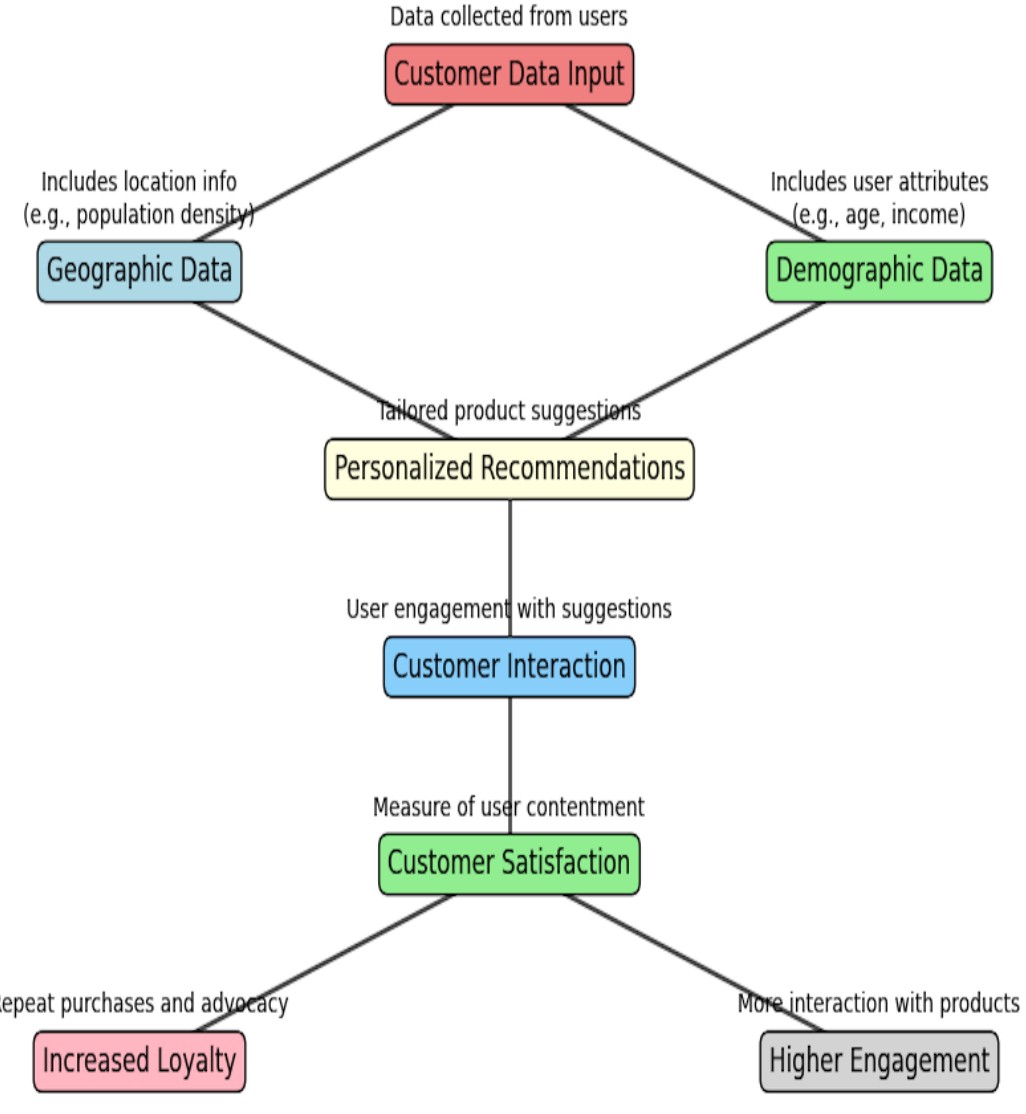

**Figure 7  Customer satisfaction and personalization.**

# WEB APPLICATION AND EXPERIMENTAL RESULTS

This section describes the infrastructure, functionality, and performance of the web application developed to meet specific user needs and business objectives. Data were obtained from The National Institute of Statistics, Tunisia (https://www.ins.tn/en).

## Web application
## Overview of the interface interactive map

Figure 10 presents the stores' position according to population when clicking the "view" button. The store offering the most products will be presented on the geographic maps.

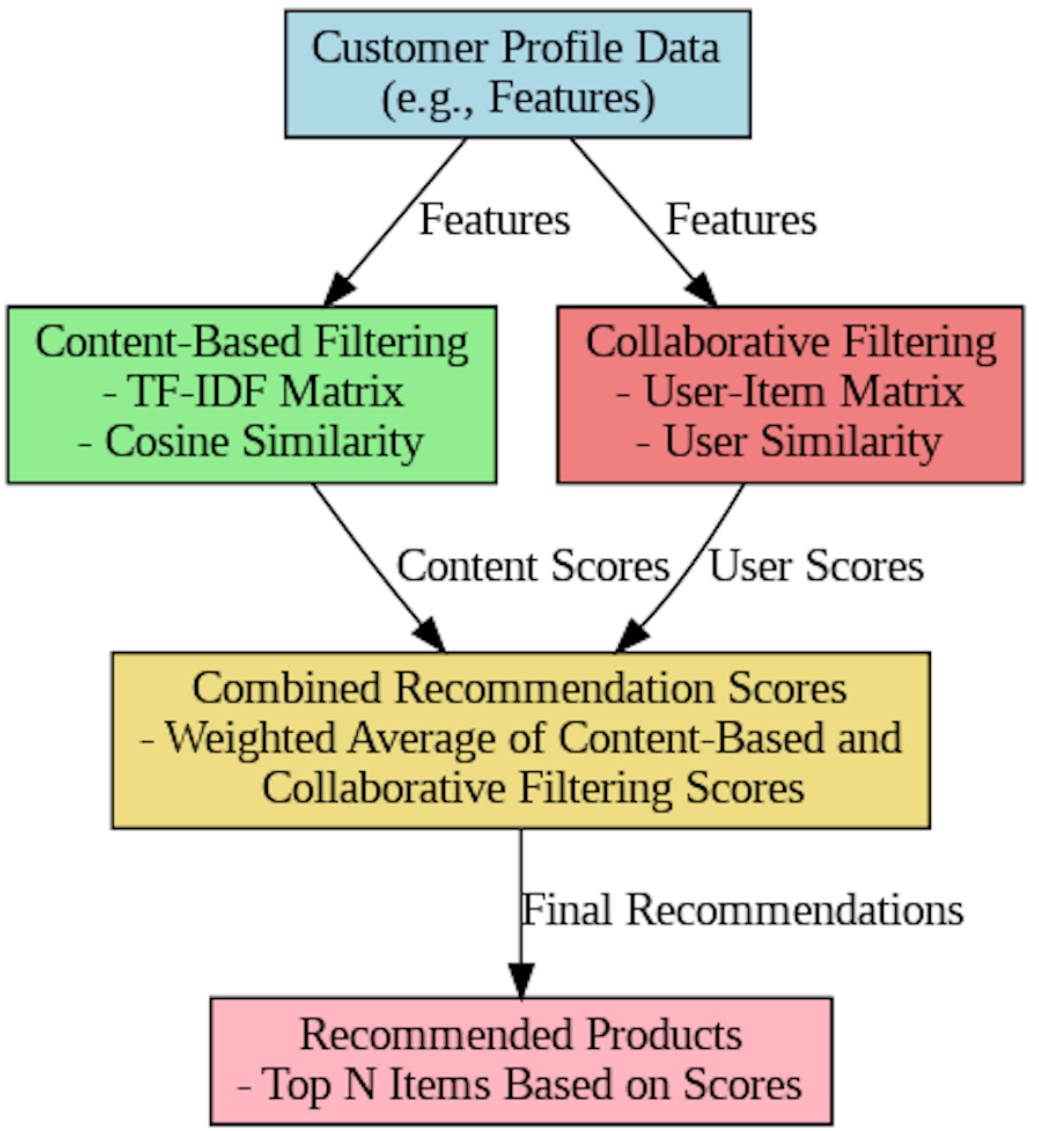

**Figure 8** Overview of the combined recommendation algorithm.

### Overview of the interface customer-product interaction analysis

Figure 11 presents the population-predicated customer product liaison analysis, and the population data are interpreted into the INS site. The coloring deterioration in this figure twists the roofing on the denseness of the population.

### Overview of dashboard

Figure 12 is an overview of the dashboard, and we offer a count of products in the store.

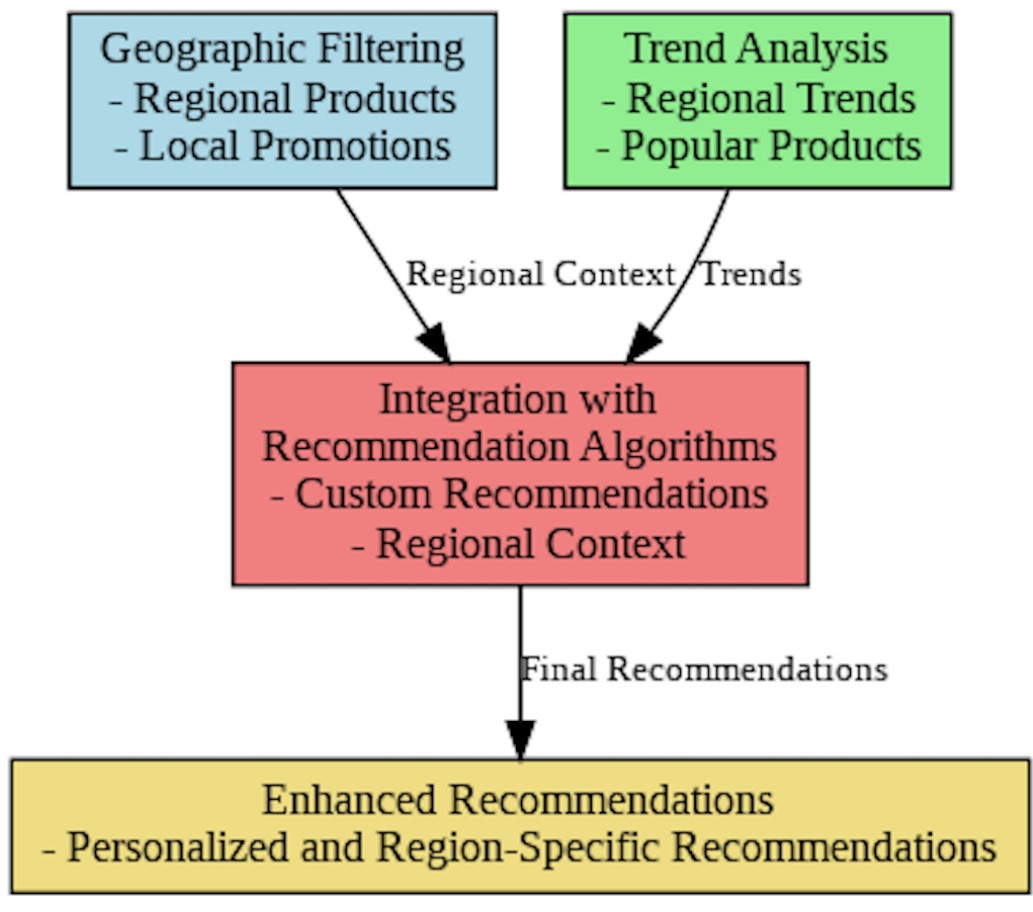

**Figure 9** Overview of the regional analysis algorithms.

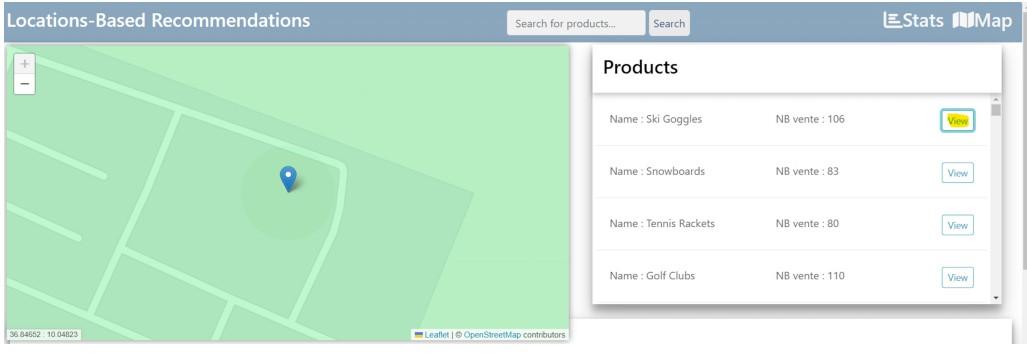

**Figure 10** The interface of the button "view".

## Product classification

Figure 13 offers interference customer behavior for each product in the store, along with the catalog of accessible products, to aid the customer in choosing the product that sparks the best benefit or is the most famous among customers.
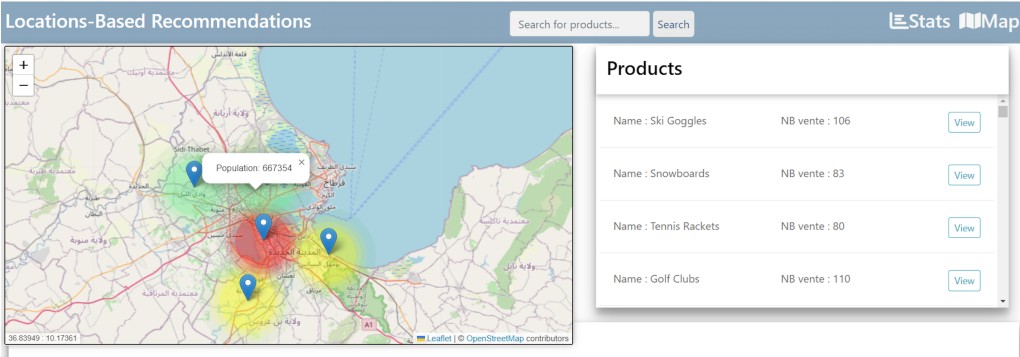

**Figure 11  Population-based customer-product interaction analysis.**

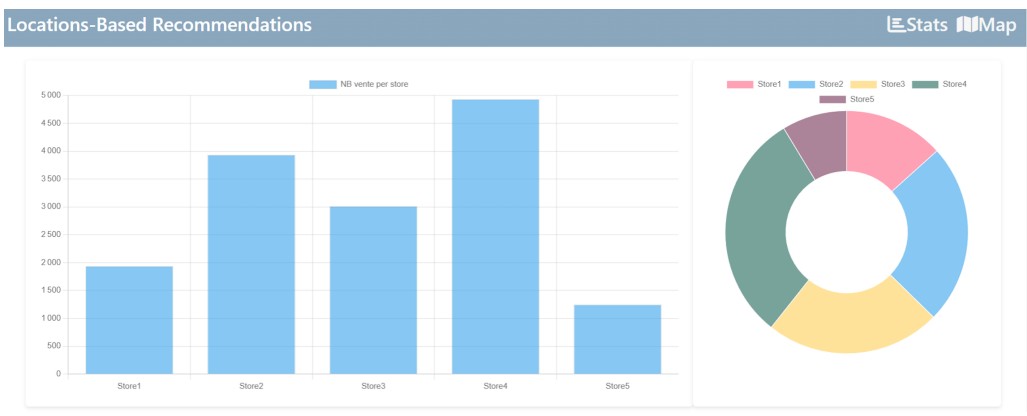

**Figure 12  Overview of dashboard.**

## Experimental results

This section focuses on the results of test and evaluating the proposed system.

### Impact of biases on predicted rating

Figure 14 demonstrates how divergence in product bias $b_y$ andconsumer bias $b_x$ influence the expected rating $\hat{R}_{xy}$ is based on 'Proposed architecture and mathematical formulation'. The expected rating handle to grow as customer and product biases boost, reflecting the composite impact of both biases, as the 3D surface figure illustrates. Higher ratings are the effect of positive customer or product biases, while lower ratings are the outcome of negative biases. This figure explains how consumer and product biases cooperate and affect the recommendation system, providing viewers with a comprehension of how modification of these elements ultimately affects the rating.

### Impact of population intensity on predicted rating

Figure 15 displays the correlation between the anticipated rating $\hat{R}_{xy}$ and population intensity. A product's population density, voiced in people per square kilometer, is

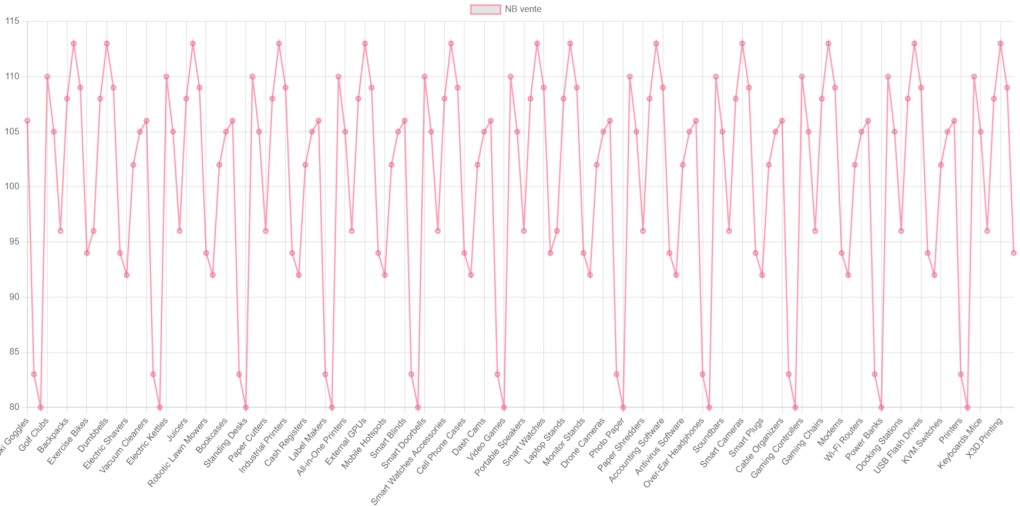

**Figure 13  Classification of products in the store.**

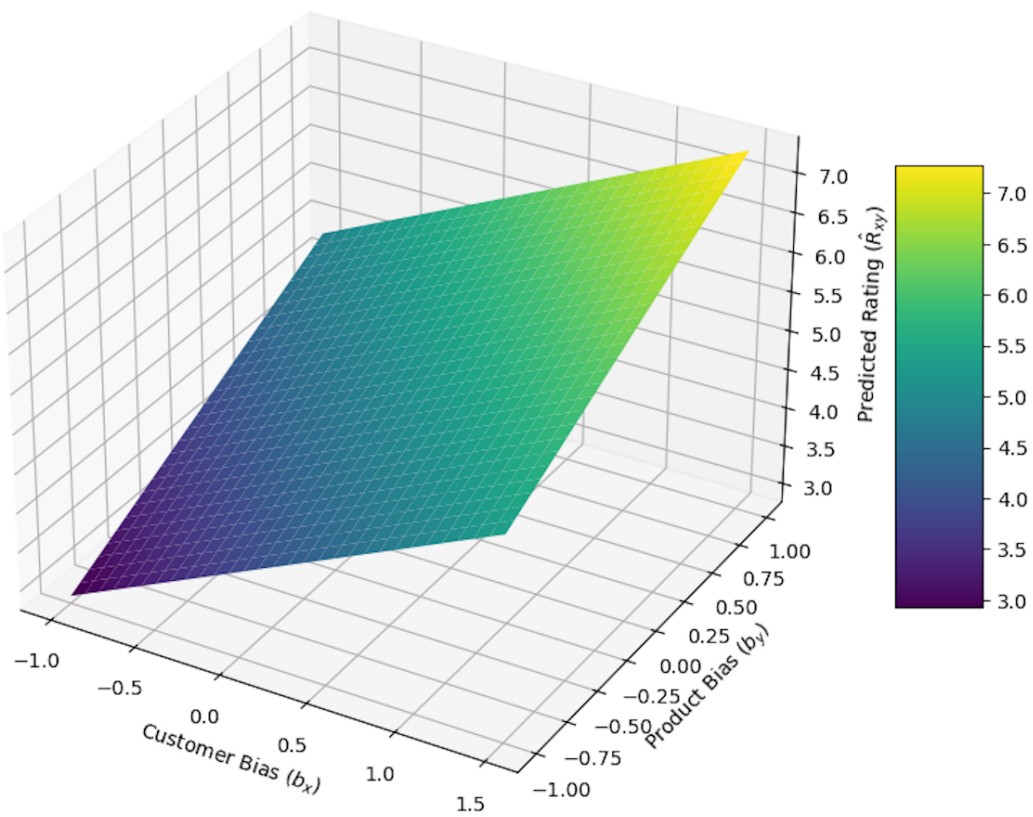

**Figure 14  Impact of customer bias and product bias on predicted rating.**

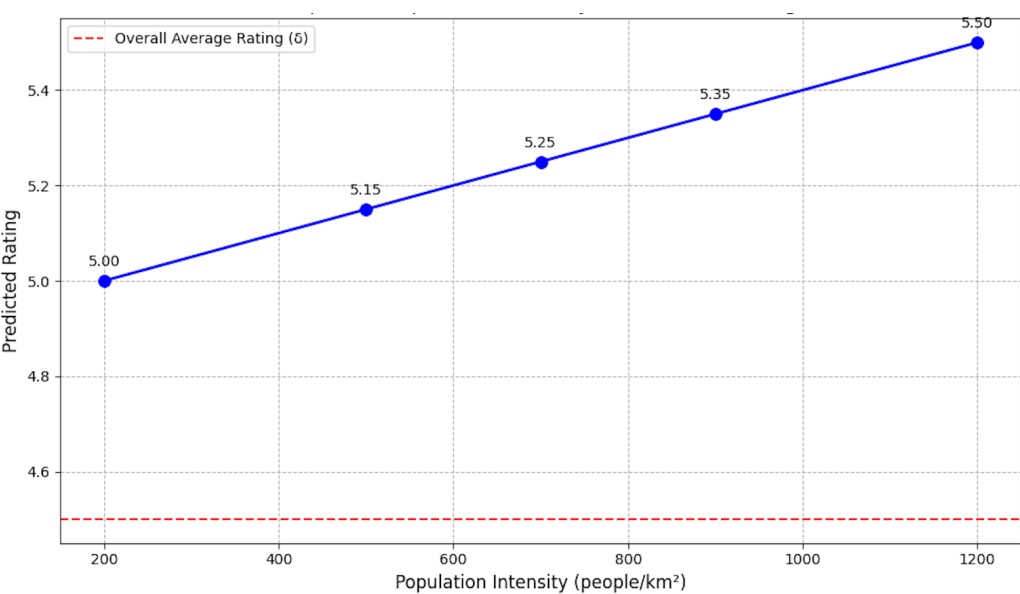

**Figure 15** Effect of population intensity on predicted rating.

accounted for by the $X$-axis, and its expected rating is displayed on the $Y$-axis. As the graph shows, enhanced population intensity results in a rise in the projected rating. This tendency results from the model's incorporation of a population intensity weight criteria $\beta$ scales with the location density of the product. The approach can enhance product recommendations in more densely populated locations; lower population intensities lead to lower anticipated ratings, whereas larger densities improve ratings. To ensure that items are assessed more positively in locations with larger population densities, this viewing demonstrates how geographical points impact the final recommendations.

## DISCUSSION

We compared our system with present-day research in demographic and geographic recommender systems to assess its effectiveness fully. Our results show that by merging demographic and geographic data to offer better accurate suggestions, our model complements and considerably boosts present methods. Geographic data is critical for upgrading suggestion importance, corresponding to research by *Nurcahya & Supriyanto (2020)* and *Addagarla & Amalanathan (2021)*. Building on these findings, our method further refines the system's capacity to customize recommendations for particular consumer groups by including demographic variables like age and income. Furthermore, integrating Geographic Information Systems (GIS) in e-commerce recommendation systems provides meaningful progress, supplying several approaches for customization and precision. For example, studies *Hussien, Rahma & Wahab (2021)* and *Kompan et al. (2022)* improve recommendations by linking them from benefit margins and contextual factors, although *Pleskach et al. (2023)* and *Yıldız, Güngör Şen & Işık (2023)* highlight personalized recommendations established on product features and visual characteristics.

This dual priority on geographic and demographic information guides to considerable advancement in performance compared to models that depend exclusively on a singular type of characteristic.

However, these models often need to pay more attention to the impact of geographic and demographic context. Our proposed geographic recommender system addresses this gap by integrating spatial data, allowing location-aware recommendations that consider user preferences and regional trends. This approach enhances the precision of recommendations and aligns with local market demands and population characteristics. Our model offers a more contextually relevant solution than traditional systems, allowing businesses to tailor recommendations based on geographic and demographic insights. This advancement leads to better-targeted marketing strategies and improved customer satisfaction, setting our approach apart from existing models that primarily focus on product attributes or user behavior without incorporating geographic context.

Nevertheless, these models repeatedly disregarded the impact of the demographic and geographic environment. By including spatial data, our proposed geographic recommender system concludes this space with favorable location-aware recommendations considering user preferences and area trends. This method enhances suggestion accuracy and fits the requirements of the local people and market. Our methodology delivers a more contextually relevant solution than previous systems, enabling organizations to personalize suggestions established on demographic and geographic data. Our method diverges from current models that primarily concentrate on product characteristics or user behavior without considering local context. This novelty resulted in better-targeted marketing maneuvers and augmented consumer satisfaction.

Through our analysis, we found that the recommender system's performance was more substantially affected by geographic factors. Because the system was successfully tailored to regional tendencies and customer requests, location-based customer preferences greatly influenced buying decisions. This completed its potential for geography data to stand out as the most important factor in upgrading the importance of product recommendations. Nevertheless, demographic variables like age and income levels also greatly influenced the system. These features were explicitly helpful in customizing product suggestions to fit the tastes and purchasing models of several clientele groups, enhancing the system's customization level.

## CONCLUSION AND FUTURE WORK

This paper presents a population-based geographic recommendation system to clarify issues established in a study in this domain incorporating crucial elements such as information collection, storage, preprocessing, analysis, visualization, and recommendation engines; the system effectively utilizes demographic and geographic data to offer highly customized recommendations. The results explain important progress in recommendation accuracy and relevance similar to those of available systems. Notably, the proposed system attains enhanced precision in aligning recommendations with user preferences and leadership to improve customer satisfaction and engagement. These

findings emphasize the system's achievable importance and its ability to progress in e-commerce. To offer better context, the system's efficiency is apparent in its capability to give better appropriate and customized recommendations, which directly translate to enhanced customer experiences. The comparative study shows clear advantages over other systems, emphasizing the novel character of this approach.

Future work could incorporate forward predictive form using artificial intelligence and machine learning. Enrichment of the customer profiles and conducive multiple criteria recommendations will enhance customization. Incorporating novel data sources, such as social media and the Internet of Things (IoT) sensors, and enhancing longer instinctive and interactive user interfaces will enhance the customer experience. This progress will offer longer, more accurate customization and be of further importance to enterprises in the e-commerce domain.

### Funding
The authors received no funding for this work.

### Competing Interests
Osama Sohaib is an Academic Editor for PeerJ Computer Science

### Author Contributions
- Mohamed Shili conceived and designed the experiments, performed the experiments, analyzed the data, performed the computation work, prepared figures and/or tables, authored or reviewed drafts of the article, and approved the final draft.
- Osama Sohaib analyzed the data, performed the computation work, prepared figures and/or tables, authored or reviewed drafts of the article, and approved the final draft.

### Data Availability
The data and source code are available in the Supplementary File.

### Supplemental Information
Supplemental information for this article can be found online at http://dx.doi.org/10.7717/peerj-cs.2525#supplemental-information.

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
