# Peer review of "Geographic recommender systems in e-commerce based on population"

_PeerJ Computer Science, doi:10.7717/peerj-cs.2525_

## Round 0.1 · original submission · Major Revisions

Dear authors,

Thank you for submitting your article. Based on reviews' comments, your article has not yet been recommended for publication in its current form. However, we encourage you to address the concerns and criticisms of the reviewers and to resubmit your article once you have updated it accordingly. Before submitting the paper, following should also be addressed:

1. Major contributions of the study should be sufficiently and appropriately written.
2. In general, the literature review is not sufficient. More recent related literature should be explored in depth. Authoritative synthesis assessing the current state-of-the-art should be provided. Advantages and disadvantages of the related works should be evaluated.
3. Generalizability of the proposed method should be mentioned.
4. Advantages and disadvantages of the methods should be clarified. Practical advantages should be indicated and the limitations of the research should be discussed.
5. The conclusion section is indicative, but it might be strengthened to highlight the importance and applicability of the work done with some more in-depth considerations, to summarize the findings, and to give readers a point of reference. Additional comments about the reached results should be included.
6. Reviewer 2 has advised you to provide specific reference. You are welcome to add it if you think it is useful and relevant. However, you are under no obligation to include it, and if you do not, it will not affect my decision.

Best wishes,

Reviewer 1 ·

Basic reporting

The paper is organized according to the structure recommended by the PeerJ Journal. However, the sections do not ensure the overall coherence of the paper. The approach in each section is relatively segregated, without connections between sections. Furthermore, the content of each section is relatively trivial and needs substantial improvements.
1. The content of the introduction section should properly present the context of the paper topic. The authors aim to demonstrate the importance of the recommender systems in e-Commerce. However, most of the introduction section addresses diverse issues (e.g., marketing optimization, logistics strategies for e-commerce businesses, user satisfaction and engagement, GIS usage, etc.) and heterogeneous references. The main contributions of the paper (lines 50 – 60) are ambitious but insufficiently revealed by the following sections.
2. The Literature review section should focus mainly on recommender systems in e-commerce. The references quoted in Section 2 are not substantially relevant to the topic of the paper. The varied papers (e.g., addressing tourist experiences, integrating GIS in logistic distribution, personalization of GIS, etc.) discussed by the authors probably contributed to the development of abilities in understanding and managing different information technologies. However, considering the topic of the paper, the literature review section should discuss and analyze studies on recommender systems in e-commerce.
Table 1 shows the heterogeneity of the quoted references and the lack of relevance to the paper topic.
References [16] and [1] are duplicated.
3. The Figures and Tables need supplementary explanations in the text. The Figures need finer, more detailed design and higher quality.
4. The Methodology section does not provide sufficient explanation to understand and assess the validity of the recommender system proposed by the authors.
5. The experimental results also need substantial supplementary clarifications.

Experimental design

The Methodology section needs substantial clarification.
1. The main characteristics of the analyzed system should precede the customer and product bias analysis (lines 155 – 169). Without appropriate definitions of the variables and parameters, the numeric example (lines 170 – 179) does not clarify the study. It is not clear whether the population intensity (line 172) is population density (in case the values should be related to square km instead of km).
2. The proposed architecture (lines 180 – 184) is presented simplistically and needs supplementary explanations.
3. The proposed approach for the recommender system (lines 185 – 193) should be described in more detail, with the presentation of the designed procedures included in the model, especially of the methods that generate recommendations based on the regional analysis (which constitute the main topic of the paper).

Validity of the findings

The presentation of the experimental results (196 – 216) lacks minimum explanations related to the developed exemplified application.
1. There is no connection to the model architecture presented in the previous section.
2. The source of data is indicated, but there are no details related to used database structures and applied procedures in data processing.
3. The results are depicted in poor-quality figures, making them difficult to understand.

Additional comments

The paper's topic is interesting. However, in my opinion, the present state of the paper should be significantly improved to meet the PeerJ requirements.
I encourage the authors to continue the research and enhance the paper's quality.

Cite this review as

Reviewer 2 ·

Basic reporting

The paper titled “Geographic recommender systems in E-Commerce based on population” explored a novel geographic recommendation system that integrating geographic and demographic data, such as population density, age, and income, to refine recommendations.

In the paper, section 3.1 can be combined with 3.2. mention the criteria for choosing the variables values. It is dependent on particular population and product data. In the figure 2, customer data is coming from population data or collected from other source. It is not clear from the flowchart. Data preprocessing step is not mentioned.

Experimental design

In this paper, geographic dataset detail is not mentioned. The used population data is from Tunisia. It is not looking as generalized recommender system. All the figures are not in good resolution quality except the figures 1& 2. In the table 2, the given layers are to be explained with respect to the data with an diagram rather than only definition.

Validity of the findings

Conclusion section is not written well for describing the key achievement and metric values. Authors can also look at the following paper for analyzing the risk factors through GIS approach.

Baruah, P., Singh, P.P., & Ojah, S.K. (2023). A Novel Framework for Risk Prediction in the Health Insurance Sector using GIS and Machine Learning. International Journal of Advanced Computer Science and Applications, 14(12), 469-476. http://dx.doi.org/10.14569/IJACSA.2023.0141249


Cite this review as

---

## Round 0.2 · Minor Revisions

Dear Authors,

Thank you for the revised paper. The original reviewers did not respond to the invitation to review the revision, so we felt it was necessary to bring in an additional reviewer. According to that reviewer, your paper still needs some minor modifications and additions. Please address the necessary concerns and criticism raised by this reviewer and resubmit the paper.

Best wishes,

Reviewer 3 ·

Basic reporting

The paper titled “Geographic Recommender Systems in E-Commerce Based on Population” presents an approach to enhance recommender systems by integrating geographic and demographic data. The Author's have incorporated the comments. To further strengthen this manuscript, there are the following suggestions:
1- The abstract should provide some more details related to the adopted methodology.

Experimental design

1- The current images appear oversized and blurry, affecting clarity.
2- The flowchart seems incomplete, requiring additional details or a more comprehensive conclusion.
After the diamond symbol, there is only a single label "No".
3- There is a need to justify the "Customer Satisfaction" feature in the proposed model.
4- What criteria guided the selection of geographic and demographic features?
5- Which feature set contributed more significantly to the recommender system's performance: demographic or geographic?

Validity of the findings

1- The results should be compared with the latest research work.

Cite this review as

---

## Round 0.3 · accepted · Accept

Dear authors,

Thank you for the revision. The paper seems sufficiently improved after the last revision and seems suitable for publication.

Best wishes,

Reviewer 3 ·

Basic reporting

It seems that the authors have incorporated the required suggestions. It can be accepted now.

Experimental design

It seems that the authors have incorporated the required suggestions. It can be accepted now.

Validity of the findings

It seems that the authors have incorporated the required suggestions. It can be accepted now.

Cite this review as